# Willingness to Pay for COVID-19 Vaccines in Japan

**DOI:** 10.3390/ijerph20227044

**Published:** 2023-11-09

**Authors:** Takeshi Yoda, Nagisa Iwasaki, Hironobu Katsuyama

**Affiliations:** 1Department of Public Health, Kawasaki Medical School, Kurashiki 701-0192, Japan; n.iwasaki@med.kawasaki-m.ac.jp (N.I.); katsu@med.kawasaki-m.ac.jp (H.K.); 2Department of Health and Sports Science, Kawasaki University of Medical Welfare, Kurashiki 701-0193, Japan

**Keywords:** willingness to pay, COVID-19, vaccine, Japan, free of charge

## Abstract

More than 80% of the Japanese population had received the coronavirus disease 2019 (COVID-19) vaccination by the end of April 2023; however, this vaccination rate continues to decline along with the need for booster shots. Further, the vaccines may not permanently be available free of charge. This study conducted a survey to determine the public’s willingness to pay for the COVID-19 vaccine in Japan. Using an internet research panel, the questionnaire collected data on various sociodemographic variables and the respondents’ willingness to pay for COVID-19 vaccines. Descriptive statistics and logistic regression analysis were used to evaluate the respondents’ answers. The results showed that of 1100 respondents, 55.2% would not want to receive the vaccine if it was paid for. A total of 44.8% respondents expressed willingness to pay, with most (170 respondents) willing to pay for 1000–1999 JPY (7.1–14.2 USD). Logistic regression analysis revealed that age, educational status, history of contracting COVID-19, and COVID-19 vaccination frequency were significantly associated with those who were willing to receive the COVID-19 vaccine if it was free (*p* < 0.05). These findings provide valuable insights for the Japanese government in determining appropriate pricing strategies to promote COVID-19 vaccination effectively.

## 1. Introduction

Since the introduction of the coronavirus diseases 2019 (COVID-19) vaccine in Japan, more than 80% of the Japanese population has been vaccinated as of the end of April 2023 [1]. Currently, three types of COVID-19 vaccines have been introduced in Japan. Two are messenger RNA (mRNA)–based vaccines BNT162b2 (Comirnaty, Pfizer, Brooklyn, NY, USA) and mRNA-1273 (Spikevax, Moderna, Cambridge, MA, USA), these vaccines were co-developed; and another one is an adjuvanted, recombinant spike protein nanoparticle vaccine (NVX-CoV2373, Novavax, Gaithersburg, MA, USA). At the time of introduction, only two types of mRNA vaccines were available. In September 2020, when the COVID-19 vaccine was in its development phase, approximately 67% of the Japanese population were willing to be vaccinated [2]. The vaccination rate has far exceeded this percentage. Factors contributing to a >80% vaccination rate may include the effectiveness of the vaccination [3,4,5] and the influence of peer pressure from individuals within one’s circle [6]. Another significant factor has been the availability of the vaccine free of charge; the COVID-19 vaccine, since its introduction, was completely free. However, the development of COVID-19 vaccines requires significant research and development funding. Hundreds of millions of dollars have been invested in companies that have successfully produced COVID-19 vaccines. The largest investments in research and development of COVID-19 vaccines were made in the United States and Germany, at approximately USD 2 billion and USD 1.5 billion, respectively. In addition, 98.12% of the approximately USD 5.9 billion in investments made until March 2021 were public funds [7]. These funding efforts have allowed for the rapid development of the COVID-19 vaccine. However, the pharmaceutical companies responsible for the development of the COVID-19 vaccine want to set the price of the vaccine in the USD 110 to USD 150 range across the board [8,9]. The COVID-19 vaccine does not currently retain immunity for several years after a single vaccination but requires additional vaccinations on a regular basis. Since the vaccination began in early 2021, a maximum of five doses have been administered to people in Japan as of February 2023.

Due to mutations in the coronavirus strain, the efficacy of the vaccine has decreased since the emergence of Delta strain [10,11]. However, the administration of additional Omicron-targeting booster shots after the third dose has been slow, despite calls from the Japanese government [1]. For the time being, Japan has stated that it will provide COVID-19 vaccines free of charge. However, “for the time being” means possibly until the end of 2023; it is hard to imagine that the vaccine will be available free of charge on a permanent basis [12]. The symptoms and transmission patterns of COVID-19 bear resemblance to those of the influenza, and similar to the COVID-19 vaccine, the influenza vaccine also requires annual administration. However, considering that a single dose of the influenza vaccine costs around USD 20, USD 110 or more per dose for the COVID-19 vaccine appears comparatively high. In Japan, there are two types of vaccinations: those that are free of charge through public subsidies, such as measles and rubella for children, and those that are completely self-funded, such as the shingles vaccine for the older people. Recently, a new type of a two-dose adjuvanted recombinant zoster vaccine (Shingrix, GSK) was made available in Japan. This recombinant vaccine currently costs about USD 150 per dose [13]. The shingles vaccine has been highly successful; vaccine efficacy against herpes zoster was 97.2% [14]. Varicella zoster virus can reactivate clinically decades after initial infection to cause herpes zoster (zoster) (i.e., shingles), a localized and generally painful cutaneous eruption that occurs most frequently among older adults. Approximately one in three persons in the general population will develop zoster during their lifetime. A common complication of zoster is postherpetic neuralgia (PHN), a chronic pain condition that can last months or even years. PHN is known to be a major cause of significantly reduced quality of life in older people [15]. In Japan, it became possible to administer live attenuated VZV vaccine in 2016, when it was approved for the purpose of preventing herpes zoster, approximately 10 years behind Europe and the United States. Nevertheless, the number of people who wish to be vaccinated against herpes zoster is low [16]. In terms of price, it is about the same as the assumed sales price of the COVID-19 vaccine, and if this amount is paid in full, out-of-pocket, the number of people who wish to be vaccinated is expected to decrease dramatically.

Therefore, we believe it is necessary to conduct a survey to determine how much the general public would be willing to pay for the COVID-19 vaccine if it were available at a fee. The results of this study should help the Japanese government determine the price that should be set to effectively promote COVID-19 vaccination.

## 2. Materials and Methods

This study was conducted in February 2023 in Japan. We used internet research panel data from QiQUMO, operated by Cross Marketing Inc., Tokyo, Japan. More than four million people were registered in the research panel. The sample size was calculated using a margin of error of 5%, a confidence level of 95%, a response distribution of 50%, and a targeted population of 110 million, giving a minimum sample size of 1067 [17,18]. Accordingly, the sample consisted of 1100 respondents, which was deemed suitable in similar previous research [19,20].

The questionnaire sought the following information: (1) sex; (2) age; (3) place of residence; (4) occupation; (5) educational background; (6) presence of chronic diseases; (7) whether the participant had ever been infected with severe acute respiratory syndrome coronavirus 2 (SARS-CoV-2); (8) presence and frequency of COVID-19 vaccination; and (9) “if the COVID-19 vaccine was paid for, how much would you be willing to pay to get it?” (I do not want to be vaccinated even if it is free; I want to be vaccinated if it is free; or I want to be vaccinated if it costs 1–999 Japanese Yen (JPY), 1000–1999 JPY, 2000–2999 JPY, 3000–3999 JPY, 4000–4999 JPY, 5000–5999 JPY, 6000–7999 JPY, 8000–9999 JPY, and more than 10,000 JPY). For reference, at the exchange rate on 20 February 2023, 100 JPY was approximately USD 0.77 and 0.71 Euros.

Descriptive statistics were used to evaluate willingness-to-pay by sex, age group, place of residence, occupation, educational background, presence of chronic diseases, experience of SARS-CoV-2 infection, and experiences of COVID-19 vaccination. We originally indicated the place of residence using prefectural levels; however, since they were too detailed to indicate any tendency, we dichotomized them as central areas (Kanto area: around Tokyo metropolis; and Kansai area: around Osaka metropolis) and others. The chi-square test was used to evaluate categorical variables, and analysis of variance (ANOVA) was used to evaluate the average age. We also analyzed the characteristics of respondents who responded that they would be willing to receive the COVID-19 vaccine if they were free using logistic regression analysis. The significance level was set at *p* < 0.05. JMP Pro 14.1.0 (SAS Institute Inc., Cary, NC, USA) was used for all analyses.

This study was approved by the Ethics Committee of Kawasaki Medical School (approval number: 5950-00). Implied rather than formal written consent was used to ensure the anonymity of the participants. The participants clicked the “I agree” button before commencing the survey to indicate their consent.

## 3. Results

Of the 1100 respondents, 547 were male (49.7%), and the average age was 46.5 years. Ages were subsequently grouped in 10-year increments for clarity during analysis. A total of 628 (57.1%) respondents lived in the central area, and 274 (24.9%) had chronic diseases. A total of 217 (19.7%) respondents had been infected with SARS-CoV-2, and 936 (85.1%) had received at least one dose of COVID-19 vaccine. The respondents’ other sociodemographic variables are shown in Table 1.

Overall, 607 (55.2%) of the 1100 respondents indicated that they would not want to receive the COVID-19 vaccine if they had to pay for one. Of them, 365 indicated that they would be willing to be vaccinated if the vaccines were free. Another 493 (44.8%) participants stated that they would be willing to receive the COVID-19 vaccine even if they were charged for it. Those who would be willing to pay for the COVID-19 vaccine were asked how much they would be willing to pay for the COVID-19 vaccine. The largest number of respondents (170) answered between 1000 JPY (≈USD 7.7) and 1999 JPY (≈USD 15.4). Only two respondents answered over 10,000 JPY (≈USD 77) for COVID-19 vaccination (Figure 1).

Respondents who did not want to be vaccinated if COVID-19 vaccine was made available for a fee were divided into two groups: those who did not want to be vaccinated even if it was free (named the “Refusal” group); and those who would be vaccinated if it was free (named the “Free” group). Further, those who responded that they would like to be vaccinated even if the COVID-19 vaccine were made available for a fee were designated as the “Willing” group, and the characteristics of each group are demonstrated in Table 2.

There were no significant differences in sex, place of residence, educational status, and previously COVDI-19 infection. However, the Willing group had a higher mean age (51.3 years old) than the other groups (43.4 and 42.2 years old, *p* < 0.01). Occupational background and previous COVID-19 vaccine frequency were also significantly reflected for the willingness to pay for the COVID-19 vaccine (*p* = 0.02 and *p* < 0.01, respectively).

Next, we performed a logistic regression analysis to identify the characteristics of those who were willing to be vaccinated if the COVID-19 vaccine was free. We used only two groups of respondents’ data, “willing to be vaccinated if it is free” and “willing to be vaccinated even if there is a fee,” and excluded the group that did not want to be vaccinated even if it is free. The reason was because the group that did not want to be vaccinated even if it was free tended to be more distrustful of the COVID-19 vaccine from the beginning [6], which was considered a barrier to identify characteristics related to willingness to pay and other factors. The results are shown in Table 3.

Logistic regression analysis revealed that age group (20–29 vs. 50–59, AOR2.26, *p* < 0.01; 20–29 vs. 60–69, AOR2.09, *p* = 0.01; 20–29 vs. 70 or more, AOR3.45, *p* < 0.01), educational status (graduate school (GS) vs. junior high school, AOR7.13, *p* < 0.01; GS vs. senior high school, AOR3.53, *p* < 0.01; GS vs. college, AOR2.62, *p* = 0.01; and GS vs. university, AOR2.29, *p* = 0.03), and COVID-19 vaccination frequency (five times vs. never, AOR2.83, *p* < 0.01; five times vs. two times, AOR1.99, *p* = 0.04; five times vs. three times, AOR2.41, *p* < 0.01; and five times vs. four times, AOR1.78, *p* = 0.01) were significant factors associated with those who indicated that they would be willing to receive the COVID-19 vaccine if it was free.

## 4. Discussion

We conducted a survey to determine how much the general public would be willing to pay for the COVID-19 vaccine if it were available at a fee, and the findings revealed that more than half of the respondents were unwilling to pay for the COVID-19 vaccine. Of these, 22% did not want to be vaccinated even if the vaccine was free. The 22% refusal is not exceptional; in many countries, approximately 20% of people refuse to be vaccinated [21,22]. Despite the widespread provision of COVID-19 vaccines and strong evidence of their efficacy [23,24], many individuals continue to express hesitancy regarding COVID-19 vaccines [20,25,26,27]. The most common reasons for this refusal were as follows: being against vaccines in general, concerns about safety or thinking that a vaccine produced in a rush is too dangerous, considering the vaccine useless because of the harmless nature of COVID-19, general lack of trust, doubts about the efficacy of the vaccine, belief in being already immunized, and doubt about the provenience of the vaccine [28]. Hesitation toward vaccines, not just the COVID-19 vaccine, is among the most important public health issues of concern for the World Health Organization (WHO) [29]. Studies have been conducted to explore factors from various perspectives regarding vaccine hesitancy, and the WHO Strategic Advisory Group of Experts (SAGE) on Immunizations defined the 3C framework, which initially spoke of vaccine confidence, complacency, and convenience [30]; later additions included calculation and collective responsibility, turning the framework into the 5C model [31]. As outlined in the model, confidence (level of trust in the safety and effectiveness of vaccines, the health system that delivers them, and the drive of policymakers who decide on the needed vaccines), complacency (low perceived risks of vaccine-preventable diseases), convenience (logistics such as availability, cost, accessibility, service quality, and uptake), calculation (active information-seeking of risks/benefits), and collective responsibility (willingness to protect others) are factors that impact vaccine hesitancy [32]. Accordingly, this study was conducted to determine the level of acceptable cost burdens in the “cost” section of the survey and found that more than half of the respondents did not like to bear the cost burden. While 20% of them are vaccine refusers who do not want to be given the vaccine even if it is free, the rest would be willing to be given the vaccine if it were free.

Even among those who indicated that they would like to receive the COVID-19 vaccine even if they had to pay for it, the overwhelming majority chose 2000 JPY or less with respect to the cost burden. This is lower than the median (28 USD) in a survey conducted in China on willingness to pay for the COVID-19 vaccine [33] but far from the USD 110–130 offered by pharmaceutical companies. Only 2 of 1100 respondents were willing to pay close to the amount proposed by the pharmaceutical company (over 10,000 JPY). This result indicated that the COVID-19 vaccination, at full payment, would be unlikely to be successful. Currently, the seasonal influenza vaccine, which is paid for and given annually to many Japanese citizens, has an immunization rate of around 35% [34]. Considering that the average cost of the influenza vaccine is USD 20–30, the desired cost of COVID-19 vaccination in this study was even lower. If the COVID-19 vaccine continues to be administered in the future, policymakers will need to continue supporting a substantial amount of vaccination costs. However, only 45% of the population still wanted to be vaccinated. To further increase the number of vaccinations, the 33% who agreed to be vaccinated if the vaccine was free will have to be included.

The characteristics of the 33% who responded that they would be willing to be vaccinated if it was free of charge were determined via logistic regression analysis, which revealed remarkable differences in the following items: younger generation, relatively lower educational level, and people who had never contracted COVID-19, never received COVID-19 vaccination, and received two or more COVID-19 vaccines. These characteristics are similar to those of respondents who hesitated to receive the COVID-19 vaccine, except with respect the item on previous COVID-19 vaccination status [6,27,35]. Vaccination campaigns for those who are hesitant to receive the COVID-19 vaccine are being conducted worldwide using various methods [36,37]. Because the group that responded that they would be willing to be vaccinated if it was free of charge had the same characteristics, we believe that the solution would be to target free vaccination to those with these characteristics. With regard to previous vaccination history with the COVID-19 vaccine, both those who had never been vaccinated at all and those who had been vaccinated two or more times were each evenly represented in terms of the characteristics of those who responded that they would be willing to be vaccinated if it were free of charge. It is odd that those who “have never been vaccinated at all” are included in the “would be willing to be vaccinated if it were free” category when the vaccination is currently available free of charge. However, it is possible that these people include those who were already infected with COVID-19 prior to vaccination. We assume that the respondents would be willing to be vaccinated if it were free in the future, because even if they were already infected, there is a possibility that they could be re-infected. The fact that significant differences were found between this group and those who had received the vaccine at least five times and those who had received the vaccine at other times suggests that making the vaccine free of charge may help to maintain vaccination to some extent. There were no significant relationships between sex, place of residence, occupation, and presence of chronic illness. A free COVID-19 vaccine would be the most effective in spreading the vaccine; however, to maximize cost-effectiveness, there may be a way to differentiate cost subsidies between those who would be willing to receive the vaccine if it were free and those who would not. For example, based on the results of this study, it is thought that offering free vaccinations to the younger generation and those who have received free COVID-19 vaccinations in the past would help maintain or improve the vaccination rate. However, it is important for preventive medicine to be cost-effective; incremental quality-adjusted life years (QALYs) gained from COVID-19 vaccination were not sought in this study. The generally acceptable threshold in Japan is less than 5–6 million JPY/QALY [38], and a health economic analysis of where the market price of this COVID-19 vaccine should be set is an important issue to be considered in the future.

This study had some limitations. First, this study was internet-based; hence, selectivity bias could not be eliminated. Second, this study was a cross-sectional study; therefore, causality could not be established. In addition, participants may have been affected by news available on the COVID-19 vaccine at the time of the survey (February 2023). Differences in respondents’ knowledge of the vaccine, particularly in terms of the effectiveness of different vaccine types and whether the vaccine is Omicron strain-compatible or not, may have influenced the results of the responses, but we did not observe this. The questionnaire was designed to be simple and easy to answer; thus, we could not evaluate sociodemographic factors other than what was conducted in this study. Furthermore, since responses to the questions were self-reported, the possibility cannot be ruled out that there may be errors, particularly with regard to previous infection with COVID-19 and the number of times the COVID-19 vaccination was administered.

Despite these limitations, our study revealed the willingness to pay for the COVID-19 vaccine among the general Japanese population. According to our findings, 45% of the Japanese population are willing to pay for the COVID-19 vaccine, but more than half of them think that an amount of less than 2000 JPY (USD 15.4) is appropriate. Although this is a survey of the Japanese public’s willingness to pay, the results of this study are expected to be similar in other COVID-19 vaccine introduction countries. Japan has a universal health coverage system, and the cost of COVID-19 treatment is kept low under this insurance system. On the other hand, vaccines for prevention are generally self-paid, and in many cases, except for children and the older people, the entire cost is paid by themselves. To maintain the COVID-19 vaccination rate, policymakers must close the monetary gap between the amount offered by vaccine manufacturers and the public’s willingness to pay.

## Figures and Tables

**Figure 1 ijerph-20-07044-f001:**
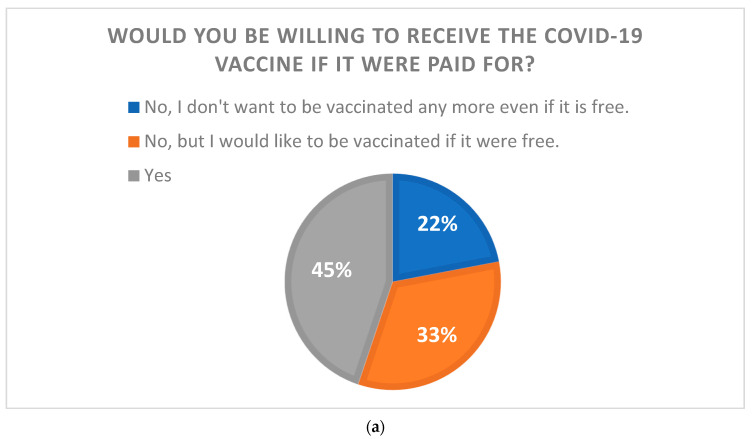
The relationship between the willingness to pay for the COVID-19 vaccine and the willingness to be vaccinated. (**a**) The results of a question asking whether respondents would be willing to be vaccinated if COVID-19 vaccine were paid for. (**b**) The results of those who were willing to pay for the vaccination when asked how much they would be willing to pay (1 JPY = USD 0.0077).

**Table 1 ijerph-20-07044-t001:** Sociodemographic variables of respondents.

		N	%
Gender	Male	547	49.7
	Female	553	50.3
Age	(Mean ± SD)	46.5 ± 16.2	
Age group	20–29	240	21.8
	30–39	208	18.9
	40–49	179	16.3
	50–59	184	16.7
	60–69	161	14.7
	70 or more	128	11.6
Place of residence	Central	628	57.1
	Others	472	42.9
Educational status	Junior High School	36	3.2
	Senior High School	305	27.7
	College	230	20.9
	University	467	42.4
	Graduate School	62	5.6
Occupation	Office worker	451	41.0
	Civil servant	50	4.5
	Self employed	29	2.6
	Healthcare worker	31	2.8
	Part-time worker	147	13.3
	Housekeeper	155	14.1
	Student	39	3.5
	Others	30	2.7
	None	168	15.3
Chronic illness	None	826	75.1
	one or more	274	24.9
COVID-19 previously infected	Yes	217	19.7
	Never	883	80.3
COVID-19 vaccination frequency	Never	164	14.9
	One dose	10	0.9
	Two doses	114	10.4
	Three doses	268	24.4
	Four doses	339	30.8
	Five doses	205	18.6

COVID-19, coronavirus disease 2019; SD, standard deviation.

**Table 2 ijerph-20-07044-t002:** Association of sociodemographic variables with willingness to pay for COVID-19 vaccine.

		Refusal (%) *	Free (%) *	Willing (%) *	*p* **
Gender	Male	115 (21.0)	174 (31.8)	258 (47.2)	0.29
	Female	127 (23.0)	191 (34.5)	235 (42.5)	
Age group	20–29	59 (24.6)	106 (44.2)	75 (31.3)	<0.01
	30–39	53 (25.5)	88 (42.3)	67 (32.2)	
	40–49	52 (29.1)	57 (31.8)	70 (39.1)	
	50–59	37 (20.1)	50 (27.2)	97 (52.7)	
	60–69	22 (13.7)	42 (26.1)	97 (60.2)	
	70 or more	19 (14.8)	22 (17.2)	87 (68.0)	
Place of residence	Central	141 (22.4)	202 (32.2)	285 (45.4)	0.71
	Others	101 (21.4)	163 (34.5)	208 (44.1)	
Educational status	Junior High School	10 (27.8)	17 (47.2)	9 (25.0)	0.06
	Senior High School	68 (22.3)	111 (36.4)	126 (41.3)	
	College	50 (21.7)	75 (32.6)	105 (45.7)	
	University	101 (21.6)	150 (32.1)	216 (46.3)	
	Graduate School	13 (20.9)	12 (19.4)	37 (59.7)	
Occupation	Office worker	99 (21.9)	154 (34.2)	198 (43.9)	0.02
	Civil servant	8 (16.0)	16 (32.0)	26 (52.0)	
	Self employed	5 (17.3)	13 (44.8)	11 (37.9)	
	Healthcare worker	2 (6.5)	17 (54.8)	12 (38.7)	
	Part-time worker	36 (24.5)	44 (29.9)	67 (45.6)	
	Housekeeper	35 (22.6)	46 (29.7)	74 (47.7)	
	Student	5 (12.8)	23 (59.0)	11 (28.2)	
	Others	7 (23.3)	9 (30.0)	14 (46.7)	
	None	45 (26.8)	43 (25.6)	80 (47.6)	
Chronic illness	None	212 (25.6)	295 (26.8)	319 (38.6)	<0.01
	one or more	30 (11.0)	70 (25.5)	174 (63.5)	
COVID-19 previously infected	Yes	54 (24.9)	67 (30.9)	96 (44.2)	0.48
	Never	188 (21.3)	298 (33.7)	397 (45.0)	
COVID-19 vaccination frequency	Never	112 (68.3)	31 (18.9)	21 (12.8)	<0.01
	One time	5 (50.0)	2 (20.0)	3 (30.0)	
	Two times	40 (35.1)	38 (33.3)	36 (31.6)	
	Three times	62 (23.1)	110 (41.1)	96 (35.8)	
	Four times	18 (5.3)	139 (41.0)	182 (53.7)	
	Five times	5 (2.4)	45 (22.0)	155 (75.6)	

* Refusal, the group that did not want to be vaccinated even if the COVID-19 vaccine was free; Free, the group that agreed to be vaccinated if the COVID-19 vaccine was free; and Willing, the group that agreed to be vaccinated even if the COVID-19 vaccine was available for a fee. COVID-19, coronavirus disease; SD, standard deviation. ** Pearson’s chi-squared test.

**Table 3 ijerph-20-07044-t003:** Results of logistic regression analysis with the group that would be willing to receive the COVID-19 vaccine if it were free as the dependent variable.

		AOR	95% CI	*p*
Gender	Male	1	-	0.37
	Female	1.16	0.828–1.636	
Age group	20–29	1	-	
	30–39	0.89	0.552–1.434	0.63
	40–49	1.47	0.886–2.439	0.13
	50–59	2.26	1.341–3.796	<0.01
	60–69	2.09	1.177–3.732	0.01
	70 or more	3.45	1.692–7.024	<0.01
Place of residence	Central	1	-	0.29
	Others	1.17	0.865–1.602	
Educational status	Junior High School	7.13	2.317–21.988	<0.01
	Senior High School	3.53	1.647–7.570	<0.01
	College	2.62	1.199–5.746	0.01
	University	2.29	1.109–4.732	0.03
	Graduate School	1	-	
Occupation	Office worker	1	-	
	Civil servant	1.14	0.559–2.334	0.71
	Self employed	2.04	0.834–5.007	0.11
	Healthcare worker	1.74	0.746–4.053	0.19
	Part-time worker	0.85	0.528–1.395	0.54
	Housekeeper	1.06	0.624–1.801	0.82
	Student	1.65	0.727–3.743	0.23
	Others	0.88	0.349–2.248	0.80
	None	1.13	0.675–1.901	0.63
Chronic illness	None	1	-	0.07
	one or more	0.71	0.495–1.032	
COVID-19 previously infected	Yes	1	-	0.05
	Never	1.46	0.993–2.144	
COVID-19 vaccination frequency	Never	2.83	1.376–5.842	<0.01
	One time	1.34	0.191–9.389	0.76
	Two times	1.99	1.024–3.870	0.04
	Three times	2.41	1.431–4.048	<0.01
	Four times	1.78	1.115–2.860	0.01
	Five times	1	-	

AOR, adjusted odds ratio; CI, confidence interval; COVID-19, coronavirus disease. The model was adjusted for sex, age, place of residence, occupation, educational background, chronic diseases, experience with COVID-19 infection, and experience with COVID-19 vaccination. Model evaluation; AICc: 1106.28, BIC: 1232.83, R^2^ = 0.102.

## Data Availability

The data presented in this study are available upon request from the corresponding author (T.Y.). The data are not publicly available due to privacy concerns.

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
