# Peer review of "Willingness to Pay for COVID-19 Vaccines in Japan"

_ijerph, 2023, doi:10.3390/ijerph20227044_

Round 1

Reviewer 1 Report

Comments and Suggestions for Authors

This manuscript was revised, adding a further parameter in the scope of the age group to make it more information. The age group (people generations) could affect the willingness to pay because of their experiences and lifestyle. You may add a further argument about each generation to the discussion to make it more informative.

I think the quality and presentation of this manuscript is acceptable.

Comments.

1. Lines 27-30. Suggest using "BNT162b2 (Comirnaty, Pfizer—BioNTech)" and "mRNA-1273 (Spikevax, Moderna—NIAID)" to make it clear because these vaccines were co-development.

2. What does it mean of the "Central" in place of the residence parameter?

Suggest clarifying it because the reader may not understand the meaning.

Is it the Honshu island, the Kanto region, or What else?

3. Lines 129-131 These sentences were stated in the US Dolla currency. Suggest adding the currency rate that you used to convert because the currency rate value is dynamic.

Author Response

Dear reviewer 1

Thank you very much for carefully reading our manuscript. We appreciate you for your helpful suggestions. We have provided point-by-point responses to your comments.

  1. Lines 27-30. Suggest using "BNT162b2 (Comirnaty, Pfizer—BioNTech)" and "mRNA-1273 (Spikevax, Moderna—NIAID)" to make it clear because these vaccines were co-development.

 Answer 1. Thank you for your suggestion. We added the sentences “these vaccines were co-developed” in the Introduction section. (Page 1, Line 29, yellow colored)

  1. What does it mean of the "Central" in place of the residence parameter?

Suggest clarifying it because the reader may not understand the meaning.

Is it the Honshu island, the Kanto region, or What else?

 Answer 2. Thank you for your opinion. We added the commentary in the Materials and Methods section. (page 3, lines 105-106, yellow colored)

  1. Lines 129-131 These sentences were stated in the US Dolla currency. Suggest adding the currency rate that you used to convert because the currency rate value is dynamic.

Answer 3. Thank you very much for your recommendation. We added the Currency conversion dates and rates in the Materials and Methods section.  (page 3, Line 99-100, yellow colored)

Reviewer 2 Report

Comments and Suggestions for Authors

General Comments:

Interesting manuscript regarding the willingness for the Japanese public to pay for COVID vaccines. The manuscript is clear, straightforward, informative and relevant. One aspect that the authors could choose to incorporate in the discussion is the general price and functioning of the Japanese public healthcare system, specifically in relation to the public healthcare costs of COVID infections. The argument in focus would be the trade-off between the costs of vaccines and the costs of providing healthcare to patients in need of care, and where the funding of these costs originates. If the costs for the care far outweigh the costs of the vaccines then an argument could be made that vaccines should be subsidized to some extent.  

Author Response

Thank  you very much for carefully reading our manuscript.

Following your suggestion, I have added a brief description of the health care system and vaccines in Japan. (Page 9, Lines 270-273)

Reviewer 3 Report

Comments and Suggestions for Authors

Thank you for the opportunity to review this manuscript. In this paper, the authors evaluate the willingness of people living in Japan to undergo COVID-19 vaccination if required to pay for the vaccines.

General comments:

While the quality of written English is vastly superior to my written Japanese, it does require careful copy-editing to bring it up to the standards of a scientific publication. There are several cases of awkward or redundant phrasing. (For example, please see lines 35-36: “Another significant factor was the vaccination being free of charge factor,” which should be rephrased as “Another significant factor has been the availability of the vaccine free of charge”, or something similar.)

Specific comments:

Line 57 – “Flu” is slang in English. Say “influenza” instead.

Line 72 – The lengthy discussion of herpes zoster (HZ) vaccination is interesting but should be shortened and moved to the discussion. Also, the authors have conflated different recommendations regarding HZ vaccination. The current adjuvanted recombinant vaccine (marketed in the US as Shingrix) is recommended for all people aged 50 years and above, not 60 and above, and it is also recommended for immunocompromised people. The older, live attenuated Oka strain vaccine (marketed as Zostavax) was recommended for ages 60 and above but considered contraindicated in many immunocompromised persons. Perhaps it would be more relevant to refer to the Japanese vaccine authority’s recommendations?

Line 97 – It may be useful here to provide the current exchange rate for international readers between Japanese yen and the US dollar, the euro, or both, just for ease of reference. (This information is provided later in the paper.)

Line 116 – What was the return rate of the survey? That is, how many surveys were sent out compared with the number of respondents?

Table 1 – The reported rate of respondents with confirmed histories of COVID-19 seems low for a survey conducted in 2023, although I appreciate that more recently-infected people may have had milder illnesses and thus not been formally diagnosed.

Figure 1 – I think it would be valuable to clarify the subgroups here more carefully. The respondents report that 22% of them would refuse COVID-19 vaccination, but this number exceeds the number of currently never-vaccinated people in the survey. This is broken down in the tables, but the figure may be misleading. “Further” or “additional” vaccine doses might be a better term, since the majority of all of these participants have already been vaccinated at least once.

Table 3 – I am a little surprised that people who have never been vaccinated were 2.83 times more likely to accept vaccinated if it were free (since it is free now, raising the question about what the current barriers for them are). What is the comparator group for the AOR? Is it the general survey population or is it other never-vaccinated people?

Comments on the Quality of English Language

As noted above, needs some work but a good first draft.

Author Response

Dear reviewer 3,

Thank you very much for carefully reading our manuscript. We appreciate you for your helpful suggestions. We have added some sentences according to your advices and have accordingly revised our manuscript. We have provided point-by-point responses to your comments.

1.General comments:

While the quality of written English is vastly superior to my written Japanese, it does require careful copy-editing to bring it up to the standards of a scientific publication. There are several cases of awkward or redundant phrasing. (For example, please see lines 35-36: “Another significant factor was the vaccination being free of charge factor,” which should be rephrased as “Another significant factor has been the availability of the vaccine free of charge”, or something similar.)

Answer 1. Thank you for your suggestion. As you indicated, we have corrected the English text. (page 1, line 36 with yellow lined)

  1. Line 57 – “Flu” is slang in English. Say “influenza” instead.

  Answer 2. Thank you for your suggestion. As you indicated, we have corrected “flu” to “influenza” (page 2, line 57).

  1. Line 72 – The lengthy discussion of herpes zoster (HZ) vaccination is interesting but should be shortened and moved to the discussion. Also, the authors have conflated different recommendations regarding HZ vaccination. The current adjuvanted recombinant vaccine (marketed in the US as Shingrix) is recommended for all people aged 50 years and above, not 60 and above, and it is also recommended for immunocompromised people. The older, live attenuated Oka strain vaccine (marketed as Zostavax) was recommended for ages 60 and above but considered contraindicated in many immunocompromised persons. Perhaps it would be more relevant to refer to the Japanese vaccine authority’s recommendations?

 Answer 3. Thank you for your opinion. We have made significant changes to this part of the text to make it make sense. (page2, lines 72-75, yellow colored.)

  1. Line 97 – It may be useful here to provide the current exchange rate for international readers between Japanese yen and the US dollar, the euro, or both, just for ease of reference. (This information is provided later in the paper.)

Answer 4. Thank you very much for your recommendation. We added the Currency conversion dates and rates in the Materials and Methods section.  (page 3, Line 99-100, yellow colored)

  1. Line 116 – What was the return rate of the survey? That is, how many surveys were sent out compared with the number of respondents?

 Answer 5. Thank you for your question. Because our survey is conducted via the Internet, all registered panelists receive an e-mail from the survey company, and those who open the e-mail are the ones who respond to the survey. The survey will close once 1100 responses have been received, so even if there are panelists who are willing to respond, they may not be able to respond. Therefore, it is not possible to calculate the response rate.

  1. Table 1 – The reported rate of respondents with confirmed histories of COVID-19 seems low for a survey conducted in 2023, although I appreciate that more recently-infected people may have had milder illnesses and thus not been formally diagnosed.

 Answer 6. Thank you for your opinion. As your pointed out, we also think that the number of people with COVID-19 existing infection is small. We also believe, as you have analyzed, that some of the more recent infections include people who have not been formally diagnosed. We have added these things in the “limitations” (page 9, lines 264-267)

  1. Figure 1 – I think it would be valuable to clarify the subgroups here more carefully. The respondents report that 22% of them would refuse COVID-19 vaccination, but this number exceeds the number of currently never-vaccinated people in the survey. This is broken down in the tables, but the figure may be misleading. “Further” or “additional” vaccine doses might be a better term, since the majority of all of these participants have already been vaccinated at least once.

  Answer 7. Thank you for your suggestion. As you pointed out, the original text could have been misleading, so we have changed the text as follows;  “No, I don't want to be vaccinated any more even if it is free.” (page4, Figure 1a, yellow colored.)

  1. Table 3 – I am a little surprised that people who have never been vaccinated were 2.83 times more likely to accept vaccinated if it were free (since it is free now, raising the question about what the current barriers for them are). What is the comparator group for the AOR? Is it the general survey population or is it other never-vaccinated people?

 Answer 8. Thank you for your opinion. We chose reference group as five times or more vaccinated people, and it is certainly odd that someone who has never been vaccinated before is 2.83 times more likely to receive the free vaccination than reference. However, it is possible that these people include those who were already infected with COVID-19 prior to vaccination. We assume that the respondents would be willing to be vaccinated if it were free in the future, because even if they were already infected, there is a possibility that they could be re-infected. We added these explanations in the Discussion section (page , lines 234-240)